# Resilience, Psychological Well-Being and Daily Functioning Following Hospitalization for Respiratory Distress Due to SARS-CoV-2 Infection

**DOI:** 10.3390/healthcare9091161

**Published:** 2021-09-04

**Authors:** Michelangelo Dini, Barbara Poletti, Sofia Tagini, Maria Rita Reitano, Elisa Allocco, Ketti Mazzocco, Gabriella Pravettoni, Bernardo Dell’Osso, Antonella D’Arminio Monforte, Stefano Centanni, Alberto Priori, Roberta Ferrucci

**Affiliations:** 1Aldo Ravelli Research Center for Neurotechnology and Experimental Brain Therapeutics, University of Milan, 20142 Milan, Italy; michelangelo.dini@unimi.it (M.D.); alberto.priori@unimi.it (A.P.); 2Department of Health Sciences (DISS), University of Milan, 20142 Milan, Italy; antonella.darminio@unimi.it (A.D.M.); stefano.centanni@unimi.it (S.C.); 3Department of Neurology and Laboratory of Neuroscience, IRCCS Istituto Auxologico Italiano, 20149 Milan, Italy; b.poletti@auxologico.it (B.P.); s.tagini@auxologico.it (S.T.); 4ASST Santi Paolo e Carlo, San Paolo University Hospital, 20142 Milan, Italy; mariella.reitano@gmail.com (M.R.R.); elisa.allocco@gmail.com (E.A.); 5Department of Oncology and Hematology-Oncology, University of Milan, 20122 Milan, Italy; ketti.mazzocco@unimi.it (K.M.); gabriella.pravettoni@unimi.it (G.P.); 6Psycho-Oncology Division, IRCCS-Istituto Europeo di Oncologia, 20141 Milan, Italy; 7Department of Biomedical and Clinical Sciences Luigi Sacco, University of Milan, 20157 Milan, Italy; bernardo.dellosso@unimi.it; 8ASST Fatebenefratelli-Sacco, 20157 Milan, Italy; 9Department of Psychiatry and Behavioural Sciences, Stanford University, Stanford, CA 94305, USA

**Keywords:** resilience, COVID-19, psychological well-being, persisting symptoms

## Abstract

The ongoing COVID-19 pandemic has affected people’s psychological well-being, and hospitalized patients could face an even greater risk of psychological distress. We aimed to study resilience in recovered COVID-19 patients after hospital discharge. We recruited 50 patients (38 males, aged 28–77) who were hospitalized for COVID-19 between March and April 2020. Participants underwent a psychological assessment 5 months after hospital discharge. We administered the Connor-Davidson Resilience Scale (CD-RISC-25), Beck’s Depression inventory-II (BDI-II), and the State-Trait Anxiety Inventory Y-form (STAI). We also evaluated the impact of persisting physical, behavioral, and cognitive symptoms on resilience. Patients reported low resilience in the months following hospital discharge (CD-RISC-25 score [mean ± SD] = 55.82 ± 20.76), compared to data from studies on the general population. Lower resilience was associated with mood disturbances in the months following clinical recovery (*p* = 0.005), persisting fatigue (*p* = 0.015), sleep changes (*p* = 0.046), and subjective cognitive complaints (*p* < 0.05). Recovered COVID-19 patients exhibit low resilience following hospital discharge, which affects psychological well-being. The presence of persisting symptoms following hospital discharge affects psychological resilience. Interventions tailored to increase resilience should be considered to improve quality of life for recovered COVID-19 patients.

## 1. Introduction

The current COVID-19 pandemic represents a significant risk for people’s psychological well-being [1]. Fear of contagion, sickness or death of relatives and close friends, job loss, financial worries, prolonged social isolation, and reduced physical activity due to nationwide lockdowns can all lead to mental health issues such as anxiety and depression [2,3,4,5,6]. Also, psychological consequences range from experiencing of survival guilt in recovered patients to Post-Traumatic Stress Disorder (PTSD) symptoms and burnout in healthcare workers [7,8,9].

Given the many unknowns about the ongoing COVID-19 pandemic and its future progression, the psychological consequences of this worldwide event are likely to be significant and long-lasting; therefore, they deserve close attention. In addition to the aforementioned stressors, hospitalized patients will have likely experienced severe acute stress linked to uncertainty regarding their health status and the immediate outcome of their hospitalization, exacerbated by the impossibility of having direct contact with their loved ones in a time of extreme need [10,11,12,13].

Individuals differ greatly in their ability to withstand, adapt and respond to difficulties and setbacks of variable entity, a personality trait known as ‘resilience’ [14]. Resilience has been shown to modulate the impact of stress on mental health status, acting as a buffer between exposure to chronic stressors and/or traumatic events and the development of internalizing psychiatric symptoms (i.e., anxiety, depression) [15,16]. Most researchers have focused on the effect of resilience on mental health following natural disasters (earthquakes, hurricanes, etc.), which usually tend to be acute events. The ongoing COVID-19 pandemic, however, represents a significantly different challenge, it being an ongoing global public health emergency for which we are not currently able to foresee an end date.

Very little is known about the effect of hospitalization for COVID-19-related respiratory distress on psychological well-being after hospital discharge, as most clinical studies have assessed the role of resilience in the recovery of patients with acute injuries (orthopaedic trauma, burns, brain injury, cardiovascular events, etc.). To the best of our knowledge, most studies to date assessing resilience in the context of COVID-19 have focused on healthcare workers and special clinical populations. Given the impact of the ongoing COVID-19 pandemic, however, we should aim towards a better understanding of psychological resilience in COVID-19 patients, as it may play a key role in predicting the functional outcomes of an ever-increasing numbers of patients worldwide.

Being hospitalized for COVID-19-related respiratory distress during the first peak of the pandemic could reasonably be considered an event capable of producing psychological distress, similarly to what has been observed in people who have faced severe injury and/or trauma. Specifically, patients will have faced great uncertainty about the short-term progression of their symptoms, in a condition of complete isolation from family members or significant others for a substantial number of days. Additionally, during their hospital stay, patients likely witnessed the death of other hospitalized patients, which could lead to the development of post-traumatic psychological symptoms. Following hospital discharge, patients might experience psychological distress caused by persisting sequelae of the disease and/or by the socio-economic consequences of the ongoing pandemic (e.g., job loss, social isolation).

The primary aim of the present study is to assess resilience and psychological well-being in adult individuals who had been hospitalized following SARS-CoV-2 infection, during the first peak (between March and April 2020) of the COVID-19 pandemic.

## 2. Materials and Methods

We recruited 50 (38 males and 12 females, aged 28–77) patients who had been hospitalized for severe SARS-CoV-2 infection in various COVID-19 units of the ASST Santi Paolo e Carlo hospitals, in Milan, Italy, between March and April 2020. Patients were recruited in the months following the resolution of acute respiratory symptoms, irrespectively of the presence of persisting physical symptoms or psychological distress. Patients were contacted from a list of recovered patients who had expressed the will to participate in COVID-19-related research projects. We administered the Montreal Cognitive Assessment (MoCA) to exclude those with dementia (defined by an adjusted score <18.28), according to published normative data [17].

Participants underwent the psychological assessment about 5 months after hospital discharge (mean ± SD = 5.33 ± 1.74 months). We gathered demographic (biological sex, age, education) and clinical data (comorbidities, duration of hospitalization, viral clearance time, type of oxygen (O_2_) therapy during hospitalization).

To evaluate resilience, we administered the Connor-Davidson Resilience Scale 25-items version (CD-RISC-25) [18]. The CD-RISC-25 contains 25 items with a Likert-type response scale (range 0–4, max. total score = 100) and assesses resilience as the sum of five factors: (i) ‘personal competence, high standards and tenacity’; (ii) ‘trust in one’s instincts’; (iii) ‘positive acceptance of change and positive relationships’; (iv) ‘control’; (v) ‘spiritual influences’ [18]. Higher scores indicate greater resilience.

We administered Beck’s Depression Inventory–II (BDI-II) [19] to assess depressive symptoms, and the State-Trait Anxiety Inventory Y-form to assess state and trait anxiety (STAI-S and STAI-T, respectively) [20]. Higher BDI-II and STAI scores indicate a greater severity of symptoms.

To evaluate self-perceived repercussions of persisting symptoms on activities of daily living, we assessed symptoms that were found to persist during the months following recovery from COVID-19 and hospital discharge (fatigue, muscular pain/weakness, sleep alterations, memory and concentration problems, tip-of-the-tongue phenomena) [21]. The severity of each symptom was assessed via a 5-point Likert-type scale (0 = None, 1 = Mild, 2 = Moderate, 3 = Severe, 4 = Very severe).

Statistical analyses were performed using IBM SPSS 25. Frequency analysis was used to analyze the distribution of categorical and ordinal data. Normality of distribution of total scores (CD-RISC-25, BDI-II, STAI-S and STAI-T) was assessed using the Kolmogorov-Smirnoff test.

CD-RISC-25 factors scores were normalized (normalized score = raw partial score/max. obtainable partial score) to explore the relevance of each individual factor in our sample. Bonferroni correction was applied to account for the issue of multiple comparisons.

Published normative data were used to categorize participants based on BDI-II score [22] and STAI scores [23].

We analyzed the effect of O_2_ therapy, depressive symptoms (BDI-II score), and persisting symptoms (fatigue, muscular pain/weakness, sleep changes, memory problems, concentration problems, tip-of-the-tongue phenomena) on resilience, using an independent samples *t*-test for dichotomous independent variables, and an ANOVA for non-dichotomous independent variables. The nonparametric equivalents were used for non-normally distributed data. Correlations were analyzed using Pearson’s *r* for normally distributed data and Spearman’s correlation coefficient (*r_s_*) for non-normally distributed data.

## 3. Results

### 3.1. Descriptive Analysis

Our sample was composed predominantly of males (76%), which likely reflects the fact that males tend to suffer from more severe COVID-19 symptoms, thus showing higher rates of hospitalization [24]. The mean age was 55.27 ± 12.07; patients were hospitalized for approximately 2 weeks (13.90 ± 9.13 days). Descriptive analysis results for demographic, clinical, and psychological variables are presented in Table 1.

During hospitalization for COVID-19, 18% of patients had received no O_2_-therapy, 54% had received low flow (via face mask) O_2_-therapy, and 26% had received high-flow O_2_-therapy via Continuous Positive Airways Pressure (CPAP) devices. 

The majority (80.43%) of participants obtained BDI-II scores ≤13, indicating absence of relevant mood disturbances; 15.22% obtained scores between 14–19, indicating mild disturbances; 4.34% obtained scores between 20–28, indicating moderate disturbances; none of the participants reported severe disturbances (BDI-II score ≥29) [19,22].

The most frequently reported symptom was fatigue (78%), followed by muscle pain and/or weakness (64%), sleep changes (64%), tip-of-the-tongue phenomena (60%), concentration problems (54%) and memory problems (52%). Detailed frequency analysis results based on symptom severity are displayed in Table 2. 

None of the participants obtained STAI-S or STAI-T scores indicative of pathological anxiety, defined by scores at least 2 SD greater than the mean, according to published normative data for the Italian population, which account for the effect of age and biological sex [23].

### 3.2. Resilience

#### 3.2.1. Total Score

CD-RISC-25 mean total score was 55.82 ± 20.76 (mean ± SD). Males obtained higher scores than females (males vs. females = 60.41 ± 19.75 vs. 41.67 ± 17.76, t[47] = 2.923, *p* = 0.005, *d* = 0.998). Resilience was positively correlated with education (*r_s_* = 0.313, *p* = 0.028), while we found no correlations with age (*r* = −0.044, *p* = 0.764), hospitalization duration (*r_s_* = 0.051, *p* = 0.727), or viral clearance time (*r* = 0.034, *p* = 0.866).

We did not find significant differences in CD-RISC-25 total scores between participants who had required different levels of O_2_-therapy during hospitalization (no O_2_-therapy vs. low-flow O_2_-therapy vs. high-flow O_2_-therapy = 63.78 ± 17.80 vs. 52.85 ± 21.76 vs. 55.28 ± 22.32, F[2,45] = 1.053, *p* = 0.357).

Participants with mood disturbances (BDI-II score > 13) obtained significantly lower CD-RISC-25 scores than those with non-significant mood disturbances (BDI-II score ≤ 13) (41.89 ± 14.77 vs. 61.42 ± 18.50, t[43] = 2.93, *p* = 0.005, *d* = 1.17).

We assessed differences in resilience between subjects who reported persistent symptoms and those who did not. CD-RISC-25 scores were lower in participants who reported fatigue (present vs. absent = 51.43 ± 19.35 vs. 67 ± 20.58, t[47] = −2.51, *p* = 0.015, *d* = 0.78), sleep changes (51.32 ± 18.25 vs. 63.55 ± 22.99, t[47] = −2.05, *p* = 0.046, *d* = 0.59), tip-of-the-tongue phenomena (48.07 ± 18.34 vs. 67.05 ± 19.20, t[47] = −3.49, *p* = 0.001, *d* = 1.01), concentration problems (48.96 ± 17.13 vs. 63.56 ± 22.11, t[47] = −2.60, *p* = 0.012, *d* = 0.74), and memory problems (49.00 ± 17.81 vs. 62.92 ± 21.57, t[47] = −2.47, *p* = 0.017, *d* = 0.70). We observed a trend between lower resilience and presence of muscle pain/weakness (51.42 ± 19.68 vs. 63.39 ± 20.92, t[47] = −2.01, *p* = 0.051, *d* = 0.59).

Persisting symptoms were not associated with age (all *p* > 0.125), sex (all *p* > 0.140), hospitalization duration (all *p* > 0.221), or viral clearance time (all *p* > 0.094). Having required O_2_-therapy during hospitalization was associated with a higher prevalence of persisting muscle pain/weakness (*p* = 0.008) and tip-of-the-tongue phenomena (*p* = 0.002).

#### 3.2.2. Factor Scores

Analysis of the five factors revealed that scores were highest in items assessing ‘positive acceptance of change and positive relationships’ (normalized score = 0.64 ± 0.23), followed by ‘Personal competence, high standards and tenacity’ (0.59 ± 0.23), ‘Control’ (0.57 ± 0.26), and ‘Trust in one’s instincts’ (0.53 ± 0.25). The lowest scores were observed in items assessing ‘Spiritual influences’ (0.30 ± 0.26).

We analyzed differences in each of the five resilience factors between patients with mood disturbances and those without. After Bonferroni correction, the presence of mood disturbances was significantly associated with lower scores in ‘Personal competence, high standards and tenacity’ (absent vs. present [mean ± SD] = 0.65 ± 0.21 vs. 0.45 ± 0.14, t[43] = 2.73, adjusted *p* = 0.045, *d* = 1.13) and ‘Trust in one’s instincts’ (0.59 ± 0.23 vs. 0.35 ± 0.17, t[43] = 2.99, adjusted *p* = 0.025, *d* = 1.22). After Bonferroni correction, there were no differences for ‘Positive acceptance of change and positive relationships’ (0.70 ± 0.20 vs. 0.51 ± 0.19, t[43] = 2.60, adjusted *p* = 0.065, *d* = 0.98), ‘Control’ (0.64 ± 0.25 vs. 0.41 ± 0.19, t[43] = 2.54, adjusted *p* = 0.075, *d* = 1.03), and ‘Spiritual influences’ (0.29 ± 0.26 vs. 0.33 ± 0.25, t[43] = −0.43, adjusted *p* = 1.000, *d* = 0.16).

We also analyzed whether persisting symptoms were associated with specific resilience factors. Descriptively, patients who reported persisting symptoms obtained lower scores in each CD-RISC-25 factor, excluding ‘Spiritual influences’ (See Table 3). After the Bonferroni correction was applied, we found significant associations between ‘Personal competence, high standards and tenacity’ and memory problems (adjusted *p* = 0.025), concentration problems (adjusted *p* = 0.035), and tip-of-the-tongue phenomena (adjusted *p* = 0.005). Lower ‘Control’ scores were associated with memory problems (adjusted *p* = 0.025), concentration problems (adjusted *p* = 0.010), and tip-of-the-tongue phenomena (adjusted *p <* 0.001). ‘Positive acceptance of change and positive relationships’ was associated only with tip-of-the-tongue phenomena (adjusted *p* = 0.024). Associations were weaker for physical and behavioral symptoms, and after Bonferroni correction was applied we only observed a significant association between ‘Control’ and muscle pain/weakness (adjusted *p* = 0.050).

## 4. Discussion

We studied resilience in a sample of adult Italian citizens who had been hospitalized during the first peak of the 2020 COVID-19 outbreak. Resilience is an adaptive psychological trait that can modulate how individuals react to challenging experiences or adversity in general by acting as a buffer against stress-related psychopathology [25].

We found that our participants exhibited low resilience compared to results from the original validation study by Connor and Davidson [18], who reported a mean total score of 80.7 for the US general population, and of 71.8 for primary care patients. These results are significantly higher than those obtained by our study, and suggest that our participants exhibited overall poor resilience. Other studies, however, have found that resilience levels in the general population might be lower than what was initially observed by Connor and Davidson [26,27]. Interestingly, a large study (N > 10,000) that used the CD-RISC-25 to measure psychological resilience in the Italian general population also observed higher scores than our sample (66.7 ± 12.4) [28]. Discrepancies between the results of different studies might be attributable to cultural differences and different demographic characteristics, as factors like biological sex, age, income, and education have all been proposed to modulate psychological resilience [29,30]. However, the demographic characteristics of our sample are very similar to those of the large sample studied by Bonaccio et al. [28], which suggests that our participants did, in fact, exhibit low resilience.

Total resilience scores observed in our sample were similar to those obtained by patients with post-traumatic stress disorder (PTSD) and subjects exposed to severe trauma. In an Italian study assessing resilience following occupational accidents, the mean CD-RISC-25 total score was found to be 61.3 ± 17.3 [31], and the original validation study by Connor and Davidson (2003) observed that mean scores of PTSD patients ranged from 47.8 to 52.8. A study assessing psychological resilience in the Chinese general population between February and March 2020 found that responders with depressive and anxiety symptoms showed CD-RISC-25 scores very similar to those observed in our study (depressed = 54.12 ± 19.27; anxious = 55.20 ± 19.81). Interestingly, they also observed significantly higher resilience in non-depressed (72.82 ± 15.01) and non-anxious (71.07 ± 16.26) responders [32]. Additionally, low resilience was a significant predictor of emotional distress and lower psychological well-being in a large sample of the Italian general population during the first peak of the 2020 COVID-19 epidemic [33].

In our study, psychological resilience was influenced by demographic variables. Specifically, males exhibited higher resilience than females, which confirms the results obtained by other studies [26,30,34]. Different resilience levels between males and females could be explained by differences in how the vasopressin and oxytocin systems regulate behavior, which suggests that perturbations of these two systems in response to external stressors might determine different consequences in males and in females [35].

We also found that higher education correlated with higher resilience, which confirms the findings of previous studies [28,36]. Having observed no differences in education between males and females in our sample, it could be argued that education represents an independent predictor of resilience. Specifically, higher education may provide patients with a better understanding of the situation, which would in turn increase resilience to external stressors. However, other variables that were not assessed in our study might also play a mediating role; namely, higher education is generally associated with higher income, which in turn has been associated with higher resilience [30,37]. This could be particularly relevant, as concerns have been raised about the socio-economic consequences of the COVID-19 pandemic and their effect on people’s physical and psychological well-being [38,39]. In addition to clinical variables, future studies will need to address the interactions between socio-economic factors and psychological well-being in recovered COVID-19 patients.

We found that lower psychological resilience was associated with the presence of self-reported, clinically relevant mood disturbance. The presence of mood disturbances following hospitalization was also specifically associated with lower scores in the ‘Personal competence, high standards and tenacity’ and ‘Trust in one’s instincts’ factors (see Figure 1). This suggests that a belief in one’s abilities and instincts could represent a specific protective factor against the development of psychological distress following highly stressful illness experiences. These findings confirm what has been observed by previous studies that have described how lower resilience is associated with increased psychological distress in response to chronic stress [40], negative life events [41], and injury or medical emergencies [42,43]. Studies have also observed that patients who have been hospitalized for COVID-19 exhibit depressive symptoms in the months following hospital discharge [21,44]. Finally, recent studies found that low resilience was associated with an increased risk of developing depression in previously hospitalized patients with COVID-19 [45].

Our results highlight the presence of persisting physical, behavioral, and cognitive symptoms in recovered COVID-19 patients, confirming the results of other recent studies [21,46,47,48]. As observed with mood alterations, the presence of persisting physical, behavioral, and cognitive symptoms was associated with lower resilience (Figure 2). Persisting symptoms are likely to impact negatively on the quality of life of recovered patients, and may delay their return to work. Indeed, our results suggest that the psychological resilience of recovered non-ICU patients is significantly affected by the presence of long-lasting symptoms, possibly even more than by the severity of respiratory symptoms in the acute phase. It is conceivable that persisting symptoms might affect resilience more than the severity of symptoms in the acute phase, given the recognized impact of chronic symptoms on psychological well-being [49,50]. In fact, while relief likely follows the resolution of potentially life-threatening acute symptoms, persisting symptoms represent a chronic stressor that patients might fail to adjust to, especially when said symptoms do not show signs of improvement.

Interestingly, lower resilience was most strongly associated with presence of tip-of-the-tongue phenomena (i.e., failure to retrieve a word, coupled with a strong feeling of knowing and with frustration at the inaccessibility of the target word) [51]. This type of complaint is often reported by patients who report subjective cognitive decline, and has also been associated with the presence of objective mild cognitive impairment [52,53,54]. In addition, we observed the presence of cognitive sequelae, in particular, affected specific resilience factors. ‘Personal competence, high standards and tenacity’ and ‘Control’ were the most affected factors, which suggests that persisting cognitive problems have a negative impact on people’s beliefs regarding their self-worth, their ability to withstand difficulties, and their level of self-control. This observation warrants further attention from future studies, as objective cognitive alterations have been reported in recovered COVID-19 patients in the months following recovery [55]. The presence of subjective cognitive deficits might represent an additional challenge, as they are often under-reported and under-diagnosed [56,57]. This could result in an ever-growing number of patients experiencing persistent mild cognitive problems, which nonetheless affect their quality of life [58], going unnoticed and untreated.

Our study presents some key limitations, in addition to those detailed so far. Firstly, the lack of a control group does not allow us to draw definitive conclusions regarding the specific impact of hospitalization on resilience, as other factors linked to the COVID-19 emergency (job loss, social isolation, physical activity reduction, sickness/death of relatives and friends) might also play a significant role. Secondly, baseline (i.e., before hospitalization) data on physical and psychological well-being were not available, therefore we cannot exclude that our sample exhibited low resilience even before hospitalization. Additionally, the sample size was small and might have caused us to ignore smaller effects, we did not assess PTSD directly, and we did not perform additional assessments at longer follow-up periods. Finally, patients who agreed to participate in the study might have decided to do so because they were experiencing persisting symptoms and/or psychological distress; therefore, the conclusions drawn from our study may not necessarily generalize to the entire population of recovered COVID-19 patients.

## 5. Conclusions

The present study shows that recovered COVID-19 patients exhibit low psychological resilience following hospital discharge; furthermore, our results suggest that the presence of persisting symptoms following hospital discharge is related to lower psychological resilience, while the severity of symptoms in the acute phase is not. As the ongoing emergency forces health systems to focus predominantly on the treatment of severe acute symptoms of COVID-19, greater attention should be focused on the quality and quantity of persisting symptoms in recovered COVID-19 patients, as they could significantly affect quality of life and slow the rate of functional recovery of discharged patients. Finally, we observed that people with mood disturbances seem to have lower psychological resilience. To sum up, our results point to a possible, unfavorable interplay between the experience of distressing events (i.e., hospitalization and persisting symptoms following hospital discharge), people’s ability to withstand, adapt and respond to such difficulties, that is psychological resilience, and individuals’ psychological well-being.

Improving psychological resilience might help to break this vicious circle. Indeed, tailored interventions have been reported to successfully increase resilience over time in both heath care professionals [59,60] and the overall population [61], especially as much as the volume of the training is high [59]. Most effective interventions include mind-fulness-based programs, cognitive-behavioral-based sessions, and the Stress Management and Resilience Training (SMART) [59,60,61]. Also, interventions might be effectively delivered both face-to-face and on-line or through specific Apps (e.g., the Provider Resilience Mobile Application—PRMA). Interestingly, on-line interventions might be especially valuable in the present moment, considering the ongoing COVID-19 pandemic [62], as studies have shown that internet cognitive-behavioral therapy (iCBT) can also be effective in improving stress-related symptoms such as sleep problems [63].

## Figures and Tables

**Figure 1 healthcare-09-01161-f001:**
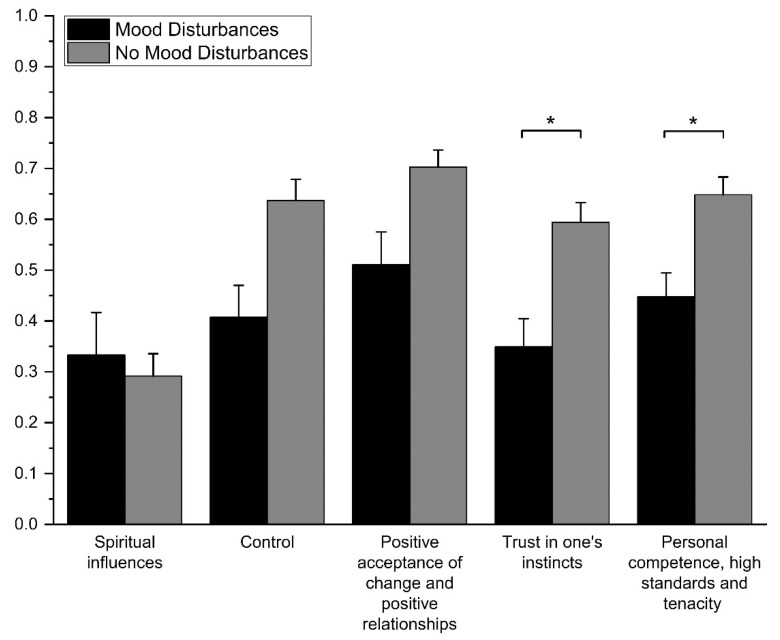
Differences in normalized CD-RISC-25 factor scores between patients with mood disturbances. CD-RISC-25 = Connor-Davidson Resilience Scale 25-items version. Asterisks denote statistically significant differences after Bonferroni correction (* = adjusted *p <* 0.05).

**Figure 2 healthcare-09-01161-f002:**
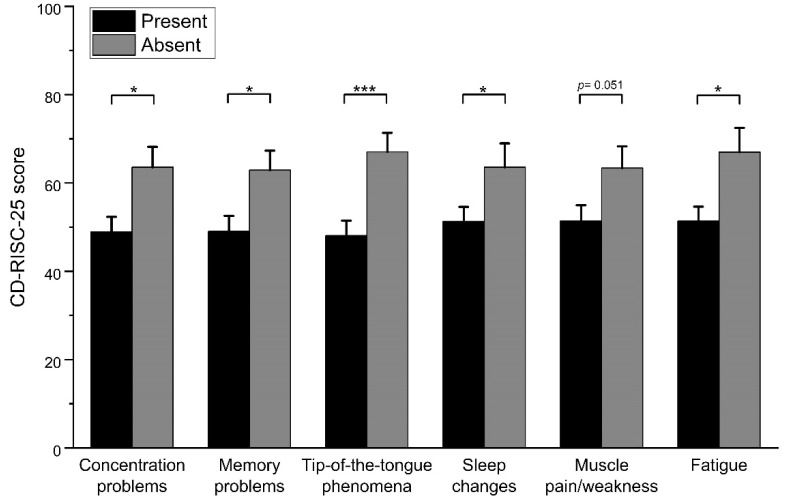
Differences in CD-RISC-25 score based on presence vs. absence of specific persisting symptoms. CD-RISC-25 = Connor-Davidson Resilience Scale 25-items version. Asterisks denote statistically significant differences (* = *p* < 0.05, *** = *p* < 0.001).

**Table 1 healthcare-09-01161-t001:** Descriptive statistics and differences between males and females.

	Females (*n* = 12)		Males (*n* = 38)		Total		
	Mean	SD	Mean	SD	Mean	SD	*p*-Value
Age	51.67	11.92	56.43	12.05	55.27	12.07	0.239
Education	11.25	3.08	13.05	3.52	12.62	3.48	0.118
Hospitalization duration (days)	11.92	9.39	14.53	9.08	13.90	9.13	0.222
Viral clearance time (days)	22.29	9.46	19.95	5.04	20.56	6.35	0.413
CD-RISC-25 score	41.67	17.76	60.41	19.75	55.82	20.76	0.005
BDI-II score	10.70	6.80	6.94	5.66	7.76	6.05	0.082
STAI-S score	35.90	9.24	32.56	8.46	33.28	8.64	0.284
STAI-T score	43.20	8.68	34.17	7.81	36.13	8.76	0.003

Data are displayed as mean and standard deviation (SD); CD-RISC-25 = Connor-Davidson Resilience Scale 25-items version; BDI-II = Beck’s Depression Inventory 2nd version; STAI = State-Trait Anxiety Inventory Y-form (-S = State anxiety; -T = Trait anxiety). *p*-values reflect the significance level of differences between males and females.

**Table 2 healthcare-09-01161-t002:** Frequency distribution of symptoms persisting after hospital discharge.

Symptom	Self-Reported Impact on Daily Functioning				
	None	Minimal	Moderate	Severe	Very Severe
Fatigue	28.00%	32.00%	26.00%	12.00%	2.00%
Muscle pain/weakness	36.00%	34.00%	22.00%	8.00%	0.00%
Sleep changes	36.00%	38.00%	20.00%	6.00%	0.00%
Tip-of-the-tongue phenomena	40.00%	32.00%	18.00%	8.00%	2.00%
Concentration problems	46.00%	32.00%	18.00%	2.00%	2.00%
Memory problems	48.00%	30.00%	14.00%	6.00%	2.00%

**Table 3 healthcare-09-01161-t003:** Differences in resilience factors based on presence of persisting symptoms.

	Fatigue		Muscle Pain/Weakness		Sleep Changes		Memory Problems		Tip-of-the-Tongue Phenomena		Concentration Problems	
	No	Yes	No	Yes	No	Yes	No	Yes	No	Yes	No	Yes
	Mean (SD)		Mean (SD)		Mean (SD)		Mean (SD)		Mean (SD)		Mean (SD)	
Personal competence, high standards and tenacity	0.70 (0.24)	0.55 (0.21)	0.65 (0.22)	0.55 (0.22)	0.66 (0.25)	0.55 (0.20)	0.68 (0.22)	0.50 (0.22)	0.72 (0.22)	0.50 (0.19)	0.68 (0.23)	0.51 (0.20)
Adjusted *p*-value	0.180		0.645		0.500		**0.025**		**0.005**		**0.035**	
Trust in one’s instincts	0.65 (0.27)	0.48 (0.23)	0.61 (0.28)	0.48 (0.22)	0.62 (0.29)	0.48 (0.21)	0.60 (0.29)	0.46 (0.18)	0.63 (0.26)	0.46 (0.22)	0.61 (0.29)	0.45 (0.18)
Adjusted *p*-value	0.155		0.475		1.000		0.215		0.075		0.165	
Positive acceptance of change, positive relationships	0.76 (0.21)	0.60 (0.22)	0.73 (0.21)	0.59 (0.22)	0.73 (0.24)	0.60 (0.21)	0.70 (0.22)	0.59 (0.22)	0.75 (0.19)	0.57 (0.22)	0.71 (0.22)	0.59 (0.22)
Adjusted *p*-value	0.115		0.155		0.255		0.360		**0.020**		0.315	
Control	0.73 (0.26)	0.51 (0.24)	0.70 (0.25)	0.50 (0.24)	0.67 (0.28)	0.52 (0.24)	0.68 (0.27)	0.47 (0.22)	0.75 (0.19)	0.46 (0.24)	0.69 (0.28)	0.47 (0.20)
Adjusted *p*-value	0.080		**0.050**		0.245		**0.025**		**0.001**		**0.010**	
Spiritual influences	0.34 (0.27)	0.29 (0.25)	0.31 (0.27)	0.30 (0.25)	0.33 (0.26)	0.29 (0.26)	0.27 (0.24)	0.34 (0.27)	0.31 (0.29)	0.30 (0.24)	0.28 (0.25)	0.32 (0.26)
Adjusted *p*-value	1.000		1.000		1.000		1.000		1.000		1.000	

Data are presented as mean and standard deviation (SD); *p*-values reflect the statistical significance (after Bonferroni correction) of differences between participants with persisting symptoms (‘Yes’ column) and those without (‘No’ column). In bold: statistically significant differences after Bonferroni correction (adjusted *p*-value < 0.05).

## Data Availability

The data that support the findings of this study are available from the corresponding author, R.F., upon reasonable request.

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
