# Peer review of "Resilience, Psychological Well-Being and Daily Functioning Following Hospitalization for Respiratory Distress Due to SARS-CoV-2 Infection"

_healthcare, 2021, doi:10.3390/healthcare9091161_

Round 1
Reviewer 1 Report
It is an important research topic that resilience, psychological well-being and daily functioning of patient with SARS-COV-2 infection, which provides profile of resilience of this kind of patient. However, small sample size limited generalization of the results of this study. Other technical issues should be addressed further as following.
1. Authors said in the section of introduction that “The primary aim of the present study is to assess the role of resilience on the post-hospitalization outcomes of adult individuals……”. It is little confusing. In fact, authors seemed to present the status of resilience and psychology of this kind of patients. Maybe authors should rewrite this sentence.
2. In the section of methods, how did the authors select the subjects? Where and how to select them? They also report severity of disease about these patients, which is helpful to understand the features of this sample for readers.
3. When authors analyzed the factors related to lower resilience, they seemed not to consider other potential confounders such as age, sex, and severity of disease. How do the authors consider this issue?
4. For table 3, authors used adjusted P value but they did not present reason why to use it in the section of methods. Possibly it is multiple-comparison problem.
Author Response
We thank the reviewer for their input. Please find below a point-by-point list of changes that were made in response to the suggested changes.
- Authors said in the section of introduction that “The primary aim of the present study is to assess the role of resilience on the post-hospitalization outcomes of adult individuals……”. It is little confusing. In fact, authors seemed to present the status of resilience and psychology of this kind of patients. Maybe authors should rewrite this sentence.
Reply: We thank the reviewer for their comment. We have rephrased the sentence to improve clarity (lines 87-89)
- In the section of methods, how did the authors select the subjects? Where and how to select them? They also report severity of disease about these patients, which is helpful to understand the features of this sample for readers.
Reply: Thank you for your comments. We have added additional details on the recruitment process (lines 97-99). More information on disease severity of the recruited patients is provided in the “descriptive analysis” section of the results (lines 142-149), which reports data on hospitalization duration, viral clearance time, and type of oxygen therapy required.
- When authors analyzed the factors related to lower resilience, they seemed not to consider other potential confounders such as age, sex, and severity of disease. How do the authors consider this issue?
Reply: We thank the reviewer for their comments. We evaluated the role of these confounders, and found that neither age nor disease severity (measured by hospitalization duration, viral clearance time, and type of oxygen therapy required) affected resilience in our sample. Sex was found to be a significant factor, with females reporting lower resilience than males. Data regarding these factors is presented at lines 170-178.
- For table 3, authors used adjusted P value but they did not present reason why to use it in the section of methods. Possibly it is multiple-comparison problem.
Reply: We thank the reviewer for their observation. We have added the rationale for the use of Bonferroni correction in the methods section (lines 128-129)
Reviewer 2 Report
Since 2020 the ongoing COVID-19 pandemic has affected all aspects of people’s living but not many published articles to specific note the outcomes of patients’ emotional and physical recovering, this paper has given us some reference to follow.
However, there is still some suggestions to make
- No IRB approval has shown in this paper, please add the participants recruitment details and ethical consideration.
- Why the female participants less than male? Please note .
- Due to all patients might discharge from different hospitals, the hospitalization experience whether might affect the outcomes of this study? please explain more about the hospital care SOP.
- The authors suggest future care plan to increase resilience intervention may useful for the COVID-19 patients, however, the resilience increasing also could be improved by the time, since the authors only use one time ( after hospital 5 months to fill all the questionnaires) to test patience’s response, whether the authors have considering to make a longer period of time to carry on the following up?
Author Response
We thank the reviewer for their input. Please find below a point-by-point list of changes that were made in response to the suggested changes.
- No IRB approval has shown in this paper, please add the participants recruitment details and ethical consideration.
Reply: We thank the reviewer for their comment. We have provided the IRB approval and ethical considerations in a dedicated paragraph (lines 384-386)
- Why the female participants less than male? Please note.
Reply: We thank the reviewer for their comment. We have added a rationale for the different proportions of male and female patients (lines 142-144).
- Due to all patients might discharge from different hospitals, the hospitalization experience whether might affect the outcomes of this study? please explain more about the hospital care SOP.
Reply: Thank you for your comment. All recruited patients were hospitalized in the ASST Santi Paolo e Carlo hospital. While patients were indeed hospitalized in different units (i.e., virology unit, respiratory unit) based on clinical needs and disease severity; these units operate under the same procedures and guidelines. Therefore, we do not expect that the hospitalization experience affected the outcomes of the study.
- The authors suggest future care plan to increase resilience intervention may useful for the COVID-19 patients, however, the resilience increasing also could be improved by the time, since the authors only use one time (after hospital 5 months to fill all the questionnaires) to test patience’s response, whether the authors have considering to make a longer period of time to carry on the following up?
Reply: We thank the reviewer for their comment. We have added the lack of longer follow-up assessments to the limitations section (lines 346-347)
Reviewer 3 Report
I have the following comments for the authors to address. I am happy to review this paper again.
1) Under the Introduction, the authors stated "Also, psychological consequences range from experiencing of survival guilt in re- 48
covered patients to Post-Traumatic Stress Disorder (PTSD) symptoms and burnout in healthcare workers [6]" Reference 6 is The Many Faces of Covid-19 at a Glance: A University Hospital Multidisciplinary Account From Milan, Italy. It is not a psychological study and does not seem to provide data to support the statement. I recommend the authors to go to Pubmed and search for the following specific findings on COVID-19 patients and healthcare workers to be included in this statement.
COVID-19 patients and please search Pubmed:
COVID-19 patients reported a higher psychological impact of the outbreak than psychiatric patients and healthy controls, with half of them having clinically significant symptoms of posttraumatic stress disorder.
Healthcare workers and please search Pubmed:
A significant association between the prevalence of physical symptoms and psychological outcomes among healthcare workers during the COVID-19 outbreak. We postulate that this association may be bi-directional, and that timely psychological interventions for healthcare workers with physical symptoms
2) The authors stated "In addition to the aforementioned stressors, hospitalized patients will have likely experienced severe acute stress
linked to uncertainty regarding their health status and the immediate outcome of their hospitalization". This statement does not have a reference.
Please go to Pubmed and search for specific findings to support this statement:
Significantly higher odds of COVID-19 hospitalization and death were found in persons with preexisting mood disorders compared with those without mood disorders
3) Under the discussion, the authors stated "Authors should discuss the results and how they can be interpreted from the perspective of previous studies and of the working hypotheses. The findings and their implications should be discussed in the broadest context possible. Future research directions may also be highlighted". I found this statement is odd. Did the authors accidentally include reviewers' comments in previous submission?
4) The authors should add one more limitation. They did not measure PTSD symptoms although they mentioned about PTSD symptoms in the discussion and introduction.
5) The authors should mention how to offer psychological treatment to COVID-19 patients. In order to reduce the risk of COVID-19 infection, the authors should propose online psychological treatment such as Internet cognitive behavior therapy (iCBT). As 64% of participants have sleep problems, please go to Pubmed to search for studies that demonstrate how iCBT can improve sleep and list this as treatment implication for this study.
Please search for the following finding on Pubmed:
Strong support for the effectiveness of dCBT-I in treating insomnia. dCBT-I has potential to revolutionise the delivery of CBT-I, improving the accessibility and availability of CBT-I content for insomnia patients worldwide
Author Response
We thank the reviewer for their input. Please find below a point-by-point list of changes that were made in response to the suggested changes.
- Under the Introduction, the authors stated "Also, psychological consequences range from experiencing of survival guilt in recovered patients to Post-Traumatic Stress Disorder (PTSD) symptoms and burnout in healthcare workers [6]" Reference 6 is The Many Faces of Covid-19 at a Glance: A University Hospital Multidisciplinary Account From Milan, Italy. It is not a psychological study and does not seem to provide data to support the statement. I recommend the authors to go to Pubmed and search for the following specific findings on COVID-19 patients and healthcare workers to be included in this statement.
COVID-19 patients and please search Pubmed:
COVID-19 patients reported a higher psychological impact of the outbreak than psychiatric patients and healthy controls, with half of them having clinically significant symptoms of posttraumatic stress disorder.
Healthcare workers and please search Pubmed:
A significant association between the prevalence of physical symptoms and psychological outcomes among healthcare workers during the COVID-19 outbreak. We postulate that this association may be bi-directional, and that timely psychological interventions for healthcare workers with physical symptoms
Reply: We thank the reviewer for their input. We have added the suggested references in the introduction (references #7-9). Reference [6] has been moved to a more relevant point and is now [1].
2) The authors stated "In addition to the aforementioned stressors, hospitalized patients will have likely experienced severe acute stress
linked to uncertainty regarding their health status and the immediate outcome of their hospitalization". This statement does not have a reference.
Please go to Pubmed and search for specific findings to support this statement:
Significantly higher odds of COVID-19 hospitalization and death were found in persons with preexisting mood disorders compared with those without mood disorders
Reply: Thank you for your comments. We have added the suggested references for the mentioned statement (ref. # 10-13).
3) Under the discussion, the authors stated "Authors should discuss the results and how they can be interpreted from the perspective of previous studies and of the working hypotheses. The findings and their implications should be discussed in the broadest context possible. Future research directions may also be highlighted". I found this statement is odd. Did the authors accidentally include reviewers' comments in previous submission?
Reply: Thank you for your observations. We have deleted the sentence, which was left from the journal’s submission template.
4) The authors should add one more limitation. They did not measure PTSD symptoms although they mentioned about PTSD symptoms in the discussion and introduction.
Reply: Thank you for your comment. We have added this limitation as per your suggestion (lines 346-347).
5) The authors should mention how to offer psychological treatment to COVID-19 patients. In order to reduce the risk of COVID-19 infection, the authors should propose online psychological treatment such as Internet cognitive behavior therapy (iCBT). As 64% of participants have sleep problems, please go to Pubmed to search for studies that demonstrate how iCBT can improve sleep and list this as treatment implication for this study.
Please search for the following finding on Pubmed:
Strong support for the effectiveness of dCBT-I in treating insomnia. dCBT-I has potential to revolutionise the delivery of CBT-I, improving the accessibility and availability of CBT-I content for insomnia patients worldwide
Reply: We thank you for your suggestion. We have added the suggested reference in the conclusions (ref. #62), thus providing a more detailed account of possible psychological interventions in these patients.
Round 2
Reviewer 1 Report
Authors have addressed most of my comments. However, the following issue may be addressed further in order to improve the mansucript.
Authors assessed differences in resilience between subjects who reported persistent symptoms and those who did not. But it is unlcear whether the other potential factors are balanced between these two groups. Authors should consider this issue in data analysis or discussion.
Author Response
Reviewer: Authors assessed differences in resilience between subjects who reported persistent symptoms and those who did not. But it is unlcear whether the other potential factors are balanced between these two groups. Authors should consider this issue in data analysis or discussion.
- Reply: We thank the reviewer for their valuable input. We have provided an analysis of other potential contributing factors between the two groups (lines 191-195). The only factor associated with any persisting symptoms was O2-therapy. It should be noted, however, that our initial analysis showed no association between resilience and O2-therapy.